# Ocular Microbiome in a Group of Clinically Healthy Horses

**DOI:** 10.3390/ani12080943

**Published:** 2022-04-07

**Authors:** Rodrigo Santibáñez, Felipe Lara, Teresa M. Barros, Elizabeth Mardones, Françoise Cuadra, Pamela Thomson

**Affiliations:** 1Departamento de Ingeniería Química y Bioprocesos, Facultad de Ingeniería, Pontificia Universidad Católica, Santiago 8940000, Chile; rlsantibanez@uc.cl; 2Unidad de Cirugía y Medicina Equina, Hospital Clínico Veterinario, Escuela de Medicina Veterinaria, Facultad de Ciencias de la Vida, Universidad Andrés Bello, Santiago 8370134, Chile; felipe.lara@unab.cl; 3Department of Clinical Science, College of Veterinary Medicine Specialty Ophthalmology Intern, Vaughan Large Animal Teaching Hospital, Auburn, AL 36832, USA; tmb0086@auburn.edu; 4Laboratorio de Microbiología Clínica y Microbioma, Escuela de Medicina Veterinaria, Facultad de Ciencias de la Vida, Universidad Andrés Bello, Santiago 8370134, Chile; elizabethmardones24@gmail.com (E.M.); francoise.cuadra@live.cl (F.C.)

**Keywords:** 16S rRNA gene, horses, microbiome, ocular surface

## Abstract

**Simple Summary:**

The microbiome of the ocular surface is composed of a large number of microorganisms dominated by bacteria and is poorly described in horses compared to other species, including humans. The objective of this study was to characterize and predict the abundance of metabolic genes of the ocular microbiome of a group of clinically healthy horses. Conjunctival swabs were obtained from both eyes of 14 horses, and DNA extraction was performed from the swabs, followed by next generation sequencing and bioinformatics analyses. The most abundant phylum was *Pseudomonadota* (*Proteobacteria*), followed by *Actinomycetota* (*Actinobacteria*) and *Bacteroidota* (*Bacteroidetes*). A total of 278 genera were identified, such as *Massilia*, *Pedobacter*, *Pseudomonas*, *Sphingomonas*, *Suttonella* and *Verticia*, among others. The inference of metabolic functions indicates that the microorganisms present in the ocular conjunctiva perform functions that point to cell growth and metabolism.

**Abstract:**

The ocular microbiome in horses is poorly described compared to other species, and most of the information available in the literature is based on traditional techniques, which has limited the depth of the knowledge on the subject. The objective of this study was to characterize and predict the metabolic pathways of the ocular microbiome of a group of healthy horses. Conjunctival swabs were obtained from both eyes of 14 horses, and DNA extraction was performed from the swabs, followed by next generation sequencing and bioinformatics analyses employing DADA2 and PICRUSt2. A total of 17 phyla were identified, of which *Pseudomonadota* (*Proteobacteria*) was the most abundant (59.88%), followed by *Actinomycetota* (*Actinobacteria*) (22.44%) and *Bacteroidota* (*Bacteroidetes*) (16.39%), totaling an average of 98.72% of the communities. Similarly, of the 278 genera identified, *Massilia*, *Pedobacter*, *Pseudomonas*, *Sphingomonas*, *Suttonella* and *Verticia* were present in more than 5% of the samples analyzed. Both *Actinobacteria* and *Bacteroides* showed great heterogeneity within the samples. The most abundant inferred metabolic functions were related to vital functions for bacteria such as aerobic respiration, amino acid, and lipid biosynthesis.

## 1. Introduction

The ocular surface is continuously exposed to the environment [1] and harbors many bacteria and other microorganisms, which influence its physiology and vary in health and disease states [2,3,4].

The tear duct represents an important physical barrier between the eye and its environment, and the surface of the eye has protective mechanisms such as the tear film and mechanical blinking that prevent the adherence and colonization of microorganisms, suggesting that only small populations of microorganisms can reside on the surface of the eye [5]. The ocular surface is not sterile, and the residing bacteria appear to have a role in the maintenance of homeostasis, modulating immune function [2] through the host production and activation of interleukins, such as IL-17 [4,6].

To study microbiome composition, massive sequencing technologies and bioinformatics analyzes have made it possible to report microbial communities with greater detail than those previously described [7,8,9,10]. For example, in horses, the most representative reported phyla residing on the ocular surface correspond to *Proteobacteria*, *Actinobacteria*, *Firmicutes* and *Bacteroidetes* [11]; furthermore, a different study identified nine phyla in the healthy conjunctiva of sampled horses. *Proteobacteria*, *Actinobacteria* and *Firmicutes*, were present in all samples analyzed, and *Proteobacteria* had the highest overall relative rate of 96.8% [12], similarly to what was described in dogs and cats [11,13]. In humans, most of the studies show a remarkable similarity to that found in animals at the phyla level, with a predominance of *Proteobacteria*, *Firmicutes* and *Actinobacteria* [5].

Changes in the composition of the microbiome during disease, known as dysbiosis, is characterized by the growth and invasion of pathogenic species [7,14,15]. Although a relationship between the microbiome and ocular pathologies has not been fully defined in animals, it is associated with some pathologies [16] such as blepharitis dysfunction of the meibomian glands, keratoconjunctivitis sicca [5], keratitis and endophthalmitis [7], Stevens–Johnson syndrome [17,18], corneal ulcer reported in human’s trauma [17]. Similarly, an experimental study in mice found an association between diabetic retinopathy and an increase in *Firmicutes* together with a decrease in *Bacteroidetes*, compared to the control group [18]. The importance of the ocular surface microbiome in health and disease has been recognized; however, there is no information related to the composition and function of the ocular microbiome in Chilean equines and international reports are scarce. In this study, we characterized the ocular microbiome in a group of healthy horses by sequencing the 16S rRNA and performing bioinformatic analyses to assign taxonomy, determine the abundance of each taxon, and to infer the abundance of metabolic functions and pathways that each microbiome harbors.

## 2. Materials and Methods

### 2.1. Ethical Approval

The study was conducted in accordance with the Declaration of Helsinki and approved by the Bioethics Committee of the Faculty of Life Sciences of the Andrés Bello University (Approval Certificate 018/2020). The study was carried out at a farm located in San Bernardo (33°37S′44.11″, 70°43′O27.44″), Metropolitan Region, Chile, during the month of November 2020, which presented average temperatures between 9.4 °C and 27.3 °C as a minimum and maximum, respectively [19].

### 2.2. Subjects and Inclusion Criteria

The animals sampled consisted of 14 adult horses regardless of age, sex, or breed which were clinically free of ocular disease [9,12]. All animals were clinically healthy and were up to date on their vaccinations and their prophylactic antiparasitic treatments. During the summer, they were kept in timothy paddocks with ad libitum feeding, while during the winter the animals were placed in stalls overnight and fed with timothy hay.

All horses waiting for their annual eye exam were sampled; the inclusion criteria considered were individuals who presented an ophthalmological examination within normal parameters and had not been under treatment with topical or systemic antibiotics at least three months before taking the sample. All horses had a complete ophthalmologic examination, consisting of an evaluation of the anterior segment of the eye with a slit-lamp biomicroscopy (SL-5, Kowa, Dan Scott and Associates, Westerville, OH, USA), and the posterior segment of the eye by indirect ophthalmoscopy (Panoptic Ophthalmoscope, Welch Allyn, Kennesaw, GA, USA). A routine ophthalmic exam was performed, including Schirmer tear test measurements (Optitech Eyecare, Med Devices Lifescience Limited, London, UK), fluoresceine sodium Dye, each strip weighing 1 mg (Optitech Eyecare, Med Devices Lifescience Limited, London, UK), and tonometry (TonoVet, iCare, Vanta, Finland). Finally, an evaluation of the pupil reflex was done with red and blue light. Any horse with an abnormal ophthalmic exam was excluded from study. Conjunctival swabs were collected before the Schirmer, fluorescein dye, and tonometry tests to prevent contamination of the sample during the examination. A volume of 0.1 mL of 0.5% tetracaine (Bausch & Lomb Inc., Tampa, FL, USA) was placed on the ocular surface of each eye to provide topical analgesia and allow for deep swabbing with applied pressure. The inferior conjunctival fornix of both eyes was sampled with Stuart swabs (Linsan, Santiago, Chile). The swab was then rubbed in the conjunctival fornix three times.

### 2.3. DNA Extraction, Targeted Sequencing and Bioinformatic Analysis

Prior to DNA extraction, each tip of the swab was cut and immersed in an Eppendorf LoBind microcentrifuge tube (Merk, Darmstadt, Alemania) with 1 mL of nuclease-free water and mixed for 5 min using a Disruptor Genie device (Scientific Industries, Bohemia, NY, USA); at this stage, negative and positive controls were used. Total DNA extraction was performed with a commercial kit (ZymoBIOMICS DNA Miniprep Kit, Zymo Research, Irvine, CA, USA) following the manufacturer instructions. The extracted DNA from each sample was diluted to 20 ng/µL in nuclease-free water (NanoDrop 2000c; Thermo Fisher Scientific, Waltham, MA, USA) and further processed at Molecular Research DNA Sequencing Services (MR-DNA, Shallowater, TX, USA). The variable region of the 16S rRNA V3-V4 gene was amplified using the primers 341F and 785R18, and the PCR products were cleaned, pooled, and sequenced in a MiSeq platform (Illumina, San Diego, CA, USA) [20].

The processed sequences were uploaded to the European Nucleotide Archive with the project code PRJEB4771. Bioinformatic analyses were performed as previously described [21] with modifications. Individual reads were processed and assigned to a bacterial taxonomy using the DADA2 v1.10 R package [22]. Briefly, the sequences were quality filtered to remove indeterminate base calls and trimmed down to 220 nucleotides. All filtered reads were used to determine an error model and to infer amplicon sequence variants (ASVs). Finally, a bacterial taxonomy was assigned to each ASV, employing a Naïve Bayesian classifier [17] and the SILVA database version 132 [23,24], and the abundance of metabolic functions and pathways were inferred for all ASV employing PICRUSt2 [25]. Rarefaction was performed with vegan v2.5-7 [26] to determine proper sequencing depth.

## 3. Results

The ocular microbiome of 14 adult, clinically healthy horses was analyzed in this study. The group consisted of 14 mares between 3 and 17 years old. Twelve of them correspond to thoroughbreds, while two of them are Chilean purebred horses, and all of them presented an ophthalmological examination within normal parameters.

Each animal was sampled twice, obtaining one swab from each eye. Swab samples were taken from the inferior conjunctival fornix. After 16S rRNA sequencing, samples of both eyes were combined and contained over 35,000 reads (62,973 ± 7546.9) and saturation, indicating that the depth of sequencing was appropriate to describe the microbial composition in these groups. Between 112 and 345 ASVs (147.07 ± 60.9) were identified from all samples (Figure 1).

Regarding all bacterial phyla identified, of a total of 17 phyla found, the most abundant and identified in all samples were *Pseudomonadota*, *Actinomycetota* and *Bacteroidota* [27,28], previously called *Proteobacteria*, *Actinobacteria*, and *Bacteroidetes*, respectively (Figure 2), which together accounted for 98.72% of the total relative abundance of bacterial communities. *Pseudomonadota* (*Proteobacteria*), represented by Gram-negative bacteria, was the most abundant phylum, with 59.88% on average for all samples (14.03% standard deviation). However, the abundance of the most abundant phyla diverged notably—in the case of *Pseudomonadota* (*Proteobacteria*), the abundance in all samples was 59.88 ± 14.30% (range 40.41–82.87%), while it varied more drastically in the case of *Actynomicetota* (*Actinobacteria*) (22.44 ± 19.13%, range 1.65–54.66%) and *Bacteroidota* (*Bacteroidetes*) (16.39 ± 18.66, range 0.58–50.46%). A total of 274 genera were identified, and the most abundant were *Pedobacter* (8.64 ± 10.45%, 0.08–25.86%), *Sphingomonas* (6.79 ± 9.85%, 0.12–29.57%), *Massilia* (5.81 ± 7.29%, 0.11–22.56%), *Pseudomonas* (5.56 ± 7.21%, 0.12–21.89%), *Verticia* (4.97 ± 9.53%, 0.03–34.17%), and *Suttonella* (4.81 ± 12.92%, 0.00–48.29%). *Suttonella* was identified in 12 of the 14 animals, and all other identified genera showed a mean relative abundance, considering all samples, lower than 5%; however, they constitute between 30.11% and 96.36% of the total composition of the microbiomes (63.41 ± 19.37%) (Figure 2).

Finally, an inference of the abundance of the metabolic content of the microbiomes was performed with the PICRUSt2 software (Figure 3). The 10 most abundant metabolic functions are related to the growth and expression of genes (DNA and RNA polymerases), aerobic metabolism (NADH: ubiquinone reductase, cytochome-c oxidase), and nutrient sensing (Histidine kinase). Meanwhile, egarding pathways, the most abundant are related to amino acid synthesis (L-isoleucine, L-valine), lipid biosynthesis (gondoate, cis-vaccenate), respiration, and nucleotide degradation (adenosine and inosine degradation). Surprisingly, PICRUSt2 predicted the presence of the engineered isobutanol pathway from pyruvate. The pathway utilizes 3-methyl-2-oxobutanoate from the L-valine biosynthesis pathway (Figure 4) and enzymes from *Lactococcus lactis* to produce isobutanol.

## 4. Discussion

The healthy ocular surface is an open system constantly exposed to the environment [1], inhabited by microorganisms that can form a stable [2,9] or transient [3] community.

The phylum that presented the highest relative abundance was *Pseudomonadota* (*Proteobacteria*); this result is consistent with previous studies carried out in other regions of the world, where the most abundant phylum found in the ocular surface of healthy horses is *Proteobacteria*, represented mainly by Gram-negative bacteria [9,29]. Interestingly, this also coincides with the microbiome that has been described on the ocular surface of humans [3,30]. However, the results obtained differ from what happens in dogs and cats, where the phylum *Firmicutes* represents the highest abundance [31,32,33]; these differences may be associated with anatomical and physiological characteristics of the eye, specific to each species. Horses have larger eyeballs that increase environmental exposure and favor the colonization of commensal or opportunistic microorganisms [12].

The most abundant genera found in this study were *Pedobacter* (8.64%), *Massilia* (5.81%), *Pseudomonas* (5.56%), among others. Of these, *Pseudomonas* has been associated with infections and the development of corneal ulcers in humans [34] and ulcerative keratitis in horses [35,36]. On the other hand, *Massilia oculi* has been reported in human subjects with endophthalmitis [37,38]. This information is important to consider, since these microorganisms have been previously associated with ocular pathology in other species.

Contrary to what this study reveals, several reports from Poland, Australia, Florida, and the United Kingdom evaluated the ocular conjunctiva in horses using culture techniques, identifying and isolating species corresponding to Gram-positive bacteria, with a predominance of *Staphylococcus* spp., *Corynebacterium* spp. or *Bacillus* spp. [39,40]. In turn, Gram-negative bacteria such as *Moraxella* spp. [41,42] *Pseudomonas* [43] and *Acinetobacter* spp. [41] were mentioned to a lesser extent. These microorganisms were found in this report, showing a relative abundance of less than 1% on average, except for *Pseudomonas*.

According to Jinks et al., 2020, in recent years, the resistance to antibiotics of different microorganisms such as *Staphylococcus*, *Streptococcus* and *Pseudomonas* isolated from horses with keratitis [43] and other pathologies has increased [44]. According to the classification made by the World Health Organization (WHO) based on the antibiotic resistance profile of some microorganisms, *Pseudomonas* and *Acinetobacter* are considered critical priorities [34,45]. It would be interesting to obtain information about the resistance genes carried by these microorganisms and their potential role in transmission dynamics.

The predicted metabolic pathways were mainly related to essential functions for bacterial life such as aerobic metabolism, cellular respiration, amino acid and lipid synthesis and nucleotide degradation [34,45,46]. There are also two components derived from fermentation processes, namely L-valine, a branched chain amino acid produced by some bacteria such as *Pseudomonas* sp. [47,48], and isobutanol, which is naturally generated by some species of the genus *Clostridium* [49]; these are metabolites that participate in metabolic exchanges between microorganisms, allowing the dynamics and stability of microbial communities [50].

## 5. Conclusions

The most abundant taxon found in this study was *Pseudomonadota* (*Proteobacteria*), which was comparable to that reported in other countries with different geographies and climates.

The prediction of metabolic routes indicates that the microorganisms present in the ocular conjunctiva of these horses perform functions that point to cell growth and metabolism, and are not related to virulence factors, which leads us to deduce that these communities are found in balance with their environment.

## Figures and Tables

**Figure 1 animals-12-00943-f001:**
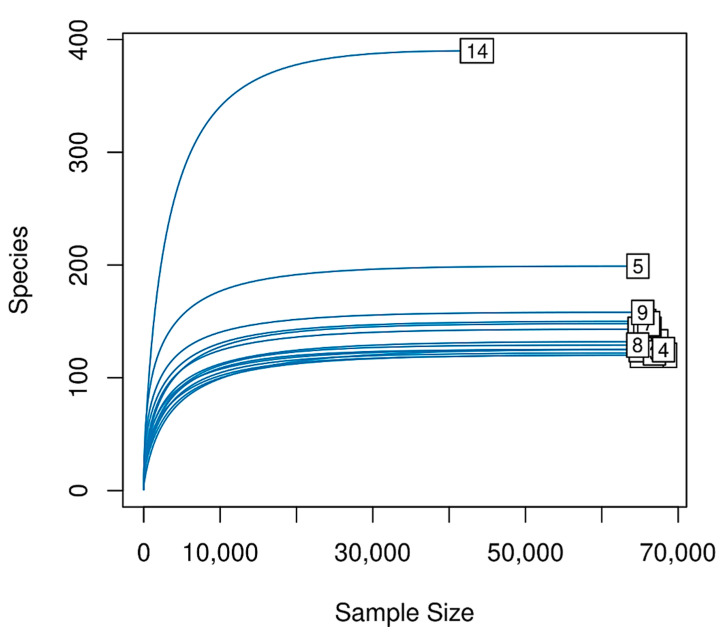
Rarefaction curves. Each sample was analyzed to determine the number of unique ASVs (*y*-axis) as a function of the sample size (*x*-axis). Rarefaction curves show saturation of the identified ASVs as the deep increased.

**Figure 2 animals-12-00943-f002:**
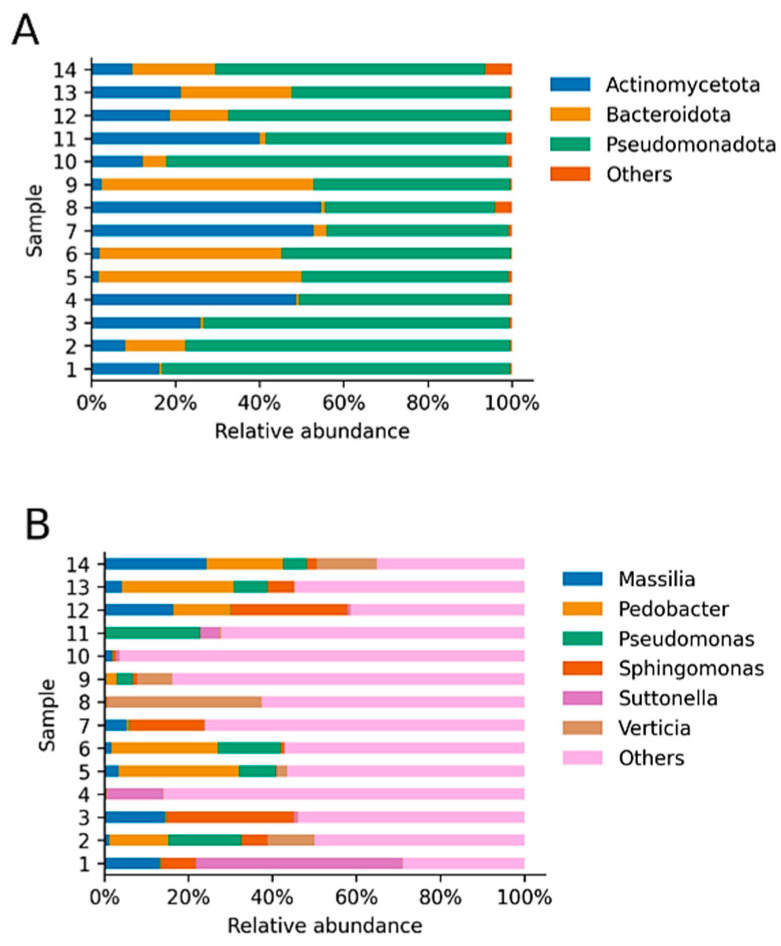
Relative abundance of taxa (**A**). Relative abundance of the most abundant identified phyla. Reads derived from both eyes were combined and phyla with an average relative abundance lower than 1% were labeled as “others”. The three plotted phyla represent 98.72% on average of the total relative abundance of all samples. (**B**). Relative abundance of the most abundant identified genera. Reads derived from both eyes were combined and genera with an average relative abundance lower than 5% were labeled as “others”. The six plotted genera represent 36.59% on average of the total relative abundance of each sample. The “other” genera represent 63.41% on average (Appendix A).

**Figure 3 animals-12-00943-f003:**
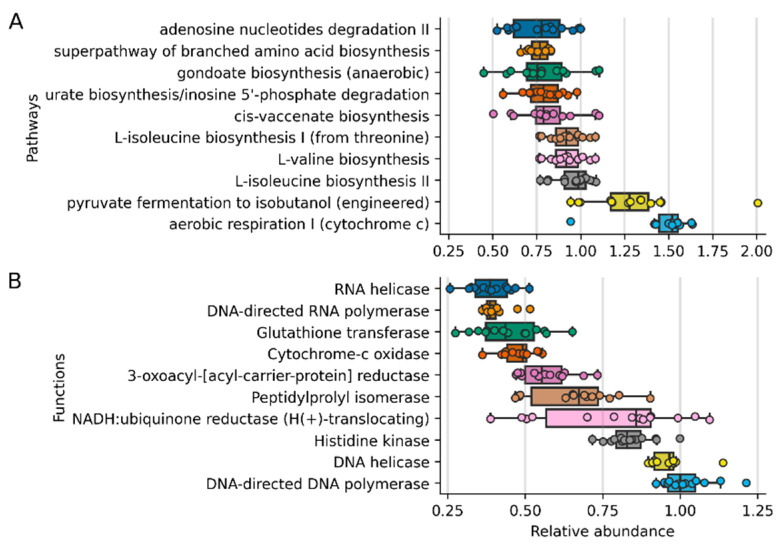
Most relatively abundant inferred pathways and functions employing PICRUSt2 (**A**). The 10 most abundant inferred pathways surpassed the 0.5% threshold. (**B**). The 10 most abundant inferred functions abundances surpassed the 0.35% threshold.

**Figure 4 animals-12-00943-f004:**
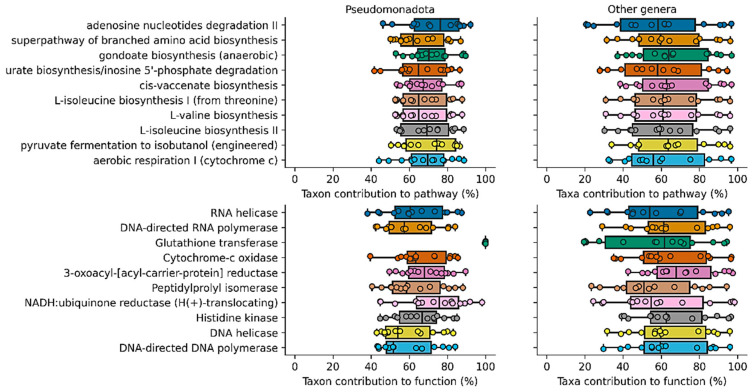
Contribution of the most abundant phylum and contribution of other genera than *Massilia*, *Pedobacter*, *Pseudomonas*, *Sphingomonas*, *Suttonella* and *Verticia* to each metabolic pathway and function per sample. *Proteobacteria* contributes more than 20% to each pathway and more than 40% to each function. In the case of genera, “others” contribute more than 20% to each pathway and function per sample.

## Data Availability

Publicly available datasets were analyzed in this study. This data can be found here https://www.ebi.ac.uk/ena/browser/view/PRJEB4771, accessed on 21 September 2021.

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
