# Peer review of "Ocular Microbiome in a Group of Clinically Healthy Horses"

_animals, 2022, doi:10.3390/ani12080943_

Round 1
Reviewer 1 Report
The authors characterized and predicted the metabolic pathways of the ocular microbiome in 14 healthy horses of various breeds. Although the study is interesting, there are some points that the authors should take into account when writing their study correctly, I am sure that it will provide the opportunity to understand it and clarify the concepts of it.
Simple summary:
Lines 16-17. The ocular surface microbiome is composed of a large catalog of microorganisms dominated by bacteria. In what species? Catalogue?.
Line 23. This study carried out in Chile describes the ocular microbiome dominated by the phylum Proteobacteria. Is repeated.
Keywords: put in alphabetical order
Introduction:
In my opinion, the review of previous studies in this area in horses has not been fully cited. For example, include history on this species of the type that appears between lines 206-212 of the discussion section.
Material and methods:
Lines 88-102: Were all the animals healthy? On what basis did the authors say they were healthy? Vaccinated, dewormed?, discipline?, stabled?, feeding regime?...
Results:
Lines 124-125: The group consisted of 13 mares and one colt, between 3 and 17 years old, of 12 thoroughbreds and 2,125 Chilean Creole horses??????. Write again. Foal was 16 years old????.
This table is not important when you have described it before.
Discussion:
Lines 206-212: In healthy eyes or in pathological eyes?.
Line 213: According to Jinks et al, 2020;. Put the reference within the animal standards.
Lines 221-226: Why is this important?
Figures:
Figure 2: What does the y-exe represent?
Figure 3 and Lines 152-153 Do the authors believe that the correlations in this study are important? Why? Each bacterium is independent of another.
Figure 4: This reviewer did not understand the meaning of this figure. What is its meaning?. What functions does each bacterium have, then?
Author Response
Reviewer 1.
Simple summary:
---Lines 16-17. The ocular surface microbiome is composed of a large catalog of microorganisms dominated by bacteria. In what species? Catalogue?.
- the word was changed
---Line 23. This study carried out in Chile describes the ocular microbiome dominated by the phylum Proteobacteria. Is repeated.
- The sentence was rewritten
----Keywords: put in alphabetical order
- Correction was made
----Introduction: In my opinion, the review of previous studies in this area in horses has not been fully cited. For example, include history on this species of the type that appears between lines 206-212 of the discussion section.
- Added a new paragraph
Material and methods: Lines 88-102: Were all the animals healthy? On what basis did the authors say they were healthy? Vaccinated, dewormed?, discipline?, stabled?, feeding regime?...
- Added a new paragraph
---Results: Lines 124-125: The group consisted of 13 mares and one colt, between 3 and 17 years old, of 12 thoroughbreds and 2,125 Chilean Creole horses??????. Write again. Foal was 16 years old????.
- The paragraph was corrected
---This table is not important when you have described it before.
- The table was deleted
---Discussion: Lines 206-212: In healthy eyes or in pathological eyes?.
- The clarification was made in the text
---Line 213: According to Jinks et al, 2020;. Put the reference within the animal standards.
- Correction was made
---Lines 221-226: Why is this important?
- The clarification was made in the text
----Figures: Figure 2: What does the y-exe represent?, Figure 3 and Lines 152-153 Do the authors believe that the correlations in this study are important? Why? Each bacterium is independent of another, Figure 4: This reviewer did not understand the meaning of this figure. What is its meaning?. What functions does each bacterium have, then?
- The figures have been modified.

Reviewer 2 Report
The manuscript by Rodrigo Santibáñez et al. describes an interesting study on the ocular microbiome of horses.
This study is well written, and deserves to be thoroughly reviewed before possible acceptance.
Global:
Prefer the passive voice
All numbers less than twelve should be written in capital letters.
Italicize bacteria names, "et al."
Specific:
The name of the journal is not "Vet.Sci" but "Animals."
Beware of the new nomenclature of bacteria names. Please check.
Line 82: include correctly this website in the references.
Methods: Authors should specify their choice of inclusion criteria (number of horses? age?). Moreover, a major problem is associated with the heterogeneity of their cohort (M/F; Breed), please homogenize, by deleting the data of these horses (n°4; 12; 14).
Table: delete the occular evaluation column, as normality is an inclusion criterion.
Lines 103-113 describe wet lab processes, not bioinformatics.
Methods: are the sample tubes low-binding tubes?
Figures 2 and 4 could be merged into one (as with Figures 5 and 6). Also, "E" could be removed because the samples are all ocular samples.
The species level is more interesting than the phyla/genera level. Give details on "other" (in the supplemental data at least).
Did the authors rarefy their data (the presentation of the rarefaction curve suggests this step, but without specifics, the question remains).
The major problem is the lack of positive and negative controls. The discussion is of great interest but could be completely wrong, in the absence of controls. The authors need to explain their methods and justify the lack of controls.
Author Response
Reviewer 2
The authors of the manuscript " Ocular microbiome in a group of clinically healthy horses" thank both reviewers for their contributions. We answer each of them below.
This study is well written and deserves to be thoroughly reviewed before possible acceptance.
Global:
---Prefer the passive voice
- Correction was made
---All numbers less than twelve should be written in capital letters.
- Correction was made
---Italicize bacteria names, "et al."
- Correction was made
---Specific: The name of the journal is not "Vet.Sci" but "Animals."
- Correction was made
---Beware of the new nomenclature of bacteria names. Please check.
- The bacterial nomenclature that we have used was the one provided by the sequencing service, we have reviewed other microbiome articles and it coincides with them. Do you have any suggestions that can guide us better?
---Line 82: include correctly this website in the references.
- Correction was made
---Methods: Authors should specify their choice of inclusion criteria (number of horses? age?). Moreover, a major problem is associated with the heterogeneity of their cohort (M/F; Breed), please homogenize, by deleting the data of these horses (n°4; 12; 14).
- Information requested by the reviewer was added to the methodology.
Regarding the heterogeneity of the participants, we have kept the 14 animals, since there are no studies that indicate that the ocular microbiome is associated with age, race or sex (unlike what can happen with the intestinal microbiome, for example). We also include a quote, where a study group like the one used in our research appears.
---Table: delete the ocular evaluation column, as normality is an inclusion criterion.
- Correction was made
---Lines 103-113 describe wet lab processes, not bioinformatics
- Correction was made
Methods: are the sample tubes low-binding tubes?
- Correction was made
---Figures 2 and 4 could be merged into one (as with Figures 5 and 6). Also, "E" could be removed because the samples are all ocular samples.
- Correction was made
---The species level is more interesting than the phyla/genera level. Give details on "other" (in the supplemental data at least)
- All genera were added with their respective relative abundance, average and standard deviation in a supplementary table.
---Did the authors rarefy their data (the presentation of the rarefaction curve suggests this step, but without specifics, the question remains)
- Correction was made
---The major problem is the lack of positive and negative controls. The discussion is of great interest but could be completely wrong, in the absence of controls. The authors need to explain their methods and justify the lack of controls
- Correction was made, the use of negative and positive controls is mentioned in the methodology.
Kind regards
Pamela Thomson M.

Round 2
Reviewer 2 Report
The manuscript by Rodrigo Santibáñez et al. has been extensively reviewed.
Some modification remains to be done mainly on taxonomy as it was recently reviewed (see for example Science depends on nomenclature, but nomenclature is not science - Karen G. Lloyd and Guillaume Tahon - https://doi.org/10.1038/
s41579-022-00684-2)
Author Response
Santiago. March 22, 2022
Dear Reviewer
We thank you again for your contributions.
The corresponding names have been modified, according to the new taxonomic proposal, for the purposes of the results mentioned by these authors. In order not to confuse the reader, the old name appears in parentheses, since in all the cited literature they are mentioned like this.
The English has been carefully reviewed by Dr. Nicolle Sallaberry, who is native to this language.
Kind regards
Pamela Thomson